# Enhanced Performance of Novel Amorphous Silicon Carbide Microelectrode Arrays in Rat Motor Cortex

**DOI:** 10.3390/mi16020113

**Published:** 2025-01-21

**Authors:** Pegah Haghighi, Eleanor N. Jeakle, Brandon S. Sturgill, Justin R. Abbott, Elysandra Solis, Veda S. Devata, Gayathri Vijayakumar, Ana G. Hernandez-Reynoso, Stuart F. Cogan, Joseph J. Pancrazio

**Affiliations:** 1Department of Bioengineering, The University of Texas at Dallas, Richardson, TX 75080, USA; pegah.haghighi@utdallas.edu (P.H.); brandon.sturgill@utdallas.edu (B.S.S.); justin.abbott@utdallas.edu (J.R.A.); elysandra.solis@utdallas.edu (E.S.); ana.hernandezreynoso@case.edu (A.G.H.-R.); sxc149830@utdallas.edu (S.F.C.); joseph.pancrazio@utdallas.edu (J.J.P.); 2Department of Chemistry and Biochemistry, The University of Texas at Dallas, Richardson, TX 75080, USA; veda.devata@utdallas.edu; 3School of Behavioral and Brain Sciences, The University of Texas at Dallas, Richardson, TX 75080, USA; gayathri.vijayakumar@utdallas.edu; 4Department of Biomedical Engineering, Case Western Reserve University, Cleveland, OH 44106, USA

**Keywords:** intracortical microelectrode arrays, neural interface, neural engineering, brain–machine interface, microelectrode

## Abstract

Implantable microelectrode arrays (MEAs) enable the recording of electrical activity from cortical neurons for applications that include brain–machine interfaces. However, MEAs show reduced recording capabilities under chronic implantation conditions. This has largely been attributed to the brain’s foreign body response, which is marked by neuroinflammation and gliosis in the immediate vicinity of the MEA implantation site. This has prompted the development of novel MEAs with either coatings or architectures that aim to reduce the tissue response. The present study examines the comparative performance of multi-shank planar, silicon-based devices and low-flexural-rigidity amorphous silicon carbide (a-SiC) MEAs that have a similar architecture but differ with respect to the shank cross-sectional area. Data from a-SiC arrays were previously reported in a prior study from our group. In a manner consistent with the prior work, larger cross-sectional area silicon-based arrays were implanted in the motor cortex of female Sprague-Dawley rats and weekly recordings were made for 16 weeks after implantation. Single unit metrics from the recordings were compared over the implantation period between the device types. Overall, the expression of single units measured from a-SiC devices was significantly higher than for silicon-based MEAs throughout the implantation period. Immunohistochemical analysis demonstrated reduced neuroinflammation and gliosis around the a-SiC MEAs compared to silicon-based devices. Our findings demonstrate that the a-SiC MEAs with a smaller shank cross-sectional area can record single unit activity with more stability and exhibit a reduced inflammatory response compared to the silicon-based device employed in this study.

## 1. Introduction

Intracortical microelectrode arrays (MEAs) enable recording of neural signals at the level of single neurons, offering insight into functional connectivity in both healthy and diseased states. Clinically, these devices have applications including, but not limited to, prostheses for restoring sensorimotor function [1] and development of neuromodulation therapies to treat neurodegenerative diseases [2]. They can be used to restore lost limb function caused by paralysis after spinal injury [3] or allow the control of prosthetic limbs after amputation [4]. Despite their potential, MEA reliability for chronic recording has remained a challenge due to signal loss over time [5,6]. Several factors have been identified as contributing to this failure mechanism including device geometry, surface topography, stiffness, and biocompatibility [7]. However, a major challenge has been the foreign body response (FBR) to the chronically implanted probe. An acute tissue response is associated with factors such as device size [8,9,10] and insertion speed [9], leading to reduced neuronal density close to the implant site [3,11]. Moreover, the chronic FBR to implanted devices is likely exacerbated by tissue–device mechanical mismatch [12] and brain micromotion [13]. As a result, macrophages and astrocytes encapsulate the devices, forming a dense barrier between viable neurons and the electrode sites [14], changing the local diffusion properties of the brain tissue that may decrease signal quality [15]. In addition, this barrier may hinder axonal regeneration, leading to neuronal death around the probe [15,16], increasing tissue impedance [17], and decreasing the extracellular spike amplitude [18].

Thus, reducing the FBR is crucial for achieving long-term reliable recording [7,19]. One approach to minimizing this response is to reduce the cross-sectional area of implanted devices. This has been linked to a demonstrable reduction in gliosis in both acute and chronic tissue response phases [20]. This led us to develop amorphous silicon carbide (a-SiC) microelectrode arrays with reduced cross-sectional areas, but that retain sufficient stiffness for intracortical implantation without external mechanical support [21]. A-SiC was selected due to its exceptional chemical stability, high electronic and ionic resistivity, biocompatibility, and suitability for thin-film fabrication processes [22,23,24]. In our prior work, we documented the chronic performance of novel a-SiC arrays [21,25]. In this work, we implanted commercially available planar, silicon-based, multi-shank MEAs and conducted an analysis of performance differences with the a-SiC arrays. Although both the a-SiC and silicon-based devices share similarities, such as having four shanks with the same length, pitch, and microelectrode site locations, a key distinction is the reduced cross-sectional area of the a-SiC probe shanks compared to that of silicon-based devices. The a-SiC MEAs have an 8 µm shank thickness with a cross-sectional area of 160 µm^2^, whereas the silicon-based probes in our study have a 15 µm shank thickness and cross-sectional area of 675 µm^2^. Due to their decreased thickness, the a-SiC devices are also more flexible (flexural rigidity = 6.4 × 10^−11^ N·m^2^ for each shank) than the silicon-based MEAs (flexural rigidity = 240 × 10^−11^ N·m^2^). Increased flexibility has been linked to stable neural recordings and minimal tissue response [26].

We performed MEA implantation and recorded unit activity over a 16-week implantation period. The presence of single units was assessed and the signal-to-noise ratio (SNR) of recordings was calculated and compared between the a-SiC and silicon-based devices. Hierarchical clustering analysis was used to classify the MEAs based on their active electrode yield (AEY) over a 16-week implantation period. Hierarchical clustering involves examining the AEY trajectories of each probe, then successively combining the two most similar trajectories until the specified number of clusters is reached [27].

Immunohistochemical characterization was performed to quantify the FBR triggered by implantation of the MEAs. Neuronal Nuclei antigen (NeuN) was used to mark neuronal cell bodies, and glial fibrillary acid protein (GFAP) identified the presence of activated astrocytes. We hypothesized that a-SiC devices would mitigate FBR under chronic implantation conditions due to their reduced cross-sectional area [28,29] and that this would be associated with an enhancement in the chronic recording performance of a-SiC microelectrode arrays. Our study highlights the enhanced chronic recording performance of a-SiC microelectrode arrays with comparatively smaller cross-sectional area shanks over otherwise similar silicon-based probes in the rat motor cortex.

## 2. Materials and Methods

### 2.1. Microelectrode Arrays

Figure 1 shows the a-SiC and silicon-based devices used in the present study. The a-SiC devices were previously characterized in [21], whereas a parallel study was performed with the silicon-based devices for comparative purposes. Each device had four collinear shanks with four channels per shank. The a-SiC MEAs were fabricated using cleanroom microfabrication techniques, as previously described [21].

For both array types, shanks were 2 mm long and designed to target layers L4/L5 of rat motor cortex. The electrode sites were spaced 200 µm apart, center-to-center, and positioned 0.7, 0.5, 0.3, and 0.1 mm from the tip of each shank. The inter-shank spacing was 200 µm center-to-center. The penetrating shanks of the a-SiC devices were 8 µm thick by 20 µm wide, giving a cross-sectional area of 160 µm^2^. These devices have previously been shown to penetrate the rat pia mater without aid of a structural support or guide [30].

Silicon-based commercial devices (A4 × 4-2 mm-200-200-200-CM16LP, NeuroNexus technologies, Ann arbor, MI, USA) had a comparable length (2 mm), shank pitch (200 µm), and electrode spacing (200 µm) to the a-SiC MEAs (160 µm^2^, 20 µm wide by 8 µm thick). However, the silicon-based devices had a larger cross-sectional area (675 µm^2^, 45 µm wide by 15 µm thick).

### 2.2. Flexural Rigidity and Stiffness Calculations

The flexural rigidity (*B*) of a-SiC and silicon-based probes was calculated using Equations (1) and (2) [31], where E represents the Young’s modulus and I the moment of inertia of the probes. The stiffness (*k_b_*) of the probes was subsequently determined using Equation (3), which accounts for the flexural rigidity and shank length (*L*).*Flexural**Rigidity* *B* = *EI*
(1)

(2)$$\mathrm{Cross-sectional}\;\mathrm{area}\;I_{rectangle}=\frac{bh^{3}}{12}$$(3)$$\mathrm{Stiffness}\;\;\;k_{b}=\frac{3EI}{L^{3}}$$

In these calculations, *L* refers to the probe’s shank length, *b* is the width of the probe, and *h* is its height. The specific values used in the calculations are summarized in Table 1.

### 2.3. MEA Implantation

All animal procedures, handling, and housing were approved by the University of Texas at Dallas Institutional Animal Care and Use Committee. Five female Sprague-Dawley rats (Charles River Laboratories, Wilmington, MA, USA) were implanted with silicon-based MEAs and compared to data previously collected using a-SiC MEAs (*n* = 7). Sprague-Dawley (SD) rats were chosen because they are a well-established model widely used in research, including motor cortex implants [34,35]. Female animals were chosen for ease of handling and because the greater aggression and hormonal fluctuations in male animals could affect outcomes. Implantation was performed following previously established procedures [36]. Anesthesia was induced and maintained in animals implanted with silicon-based MEAs using 2–2.5% isoflurane supplemented with pure O_2_ (500 mL/min). Ophthalmic ointment was applied to the eyes. The scalp was shaved to expose the skin, and anesthesia was confirmed by toe-pinch. The animal’s scalp was sterilized with alternating 10% povidone-iodine solution and 70% alcohol followed by administration of 0.16 mL of 2% lidocaine (Covetrus Inc., Portland, ME, USA). An incision was made and tissue resected to expose the skull. Quadrants were defined based on the bregma and suture lines. Three anchoring stainless steel bone screws were placed in the quadrants adjacent to the implant set. A 2 mm by 2 mm craniotomy was made over the left motor cortex 2 mm anterior to the bregma and lateral to the suture line. The motor cortex was selected for recording because of its relevance to the development of brain–machine interfacing devices. It is also known to exhibit neural activity in anesthetized animals [37]. The dura mater was resected and stainless-steel ground and reference wires from the device were wrapped around the anchoring screws. MEAs were implanted using an electronically controlled micro-positioner (NeuralGlider, Actuated Medical, Inc., Ann Arbor, MI, USA) at 100 µm/s with ultrasonic actuation to an approximate depth of 1.5 mm from the cortical surface so that all electrode sites were located within L4 and L5 of the primary motor cortex. A silicone elastomer (Kwik-Sil, World Precision Instruments, Sarasota, FL, USA) was used to fill the craniotomy and allowed to cure over several minutes. Cold-cure dental cement (A-M Systems Inc, Sequim, WA, USA) was applied to the skull to create a headcap, partially encapsulating the MEA, bone screws, and extracranial ground and reference wires. The incision site was closed with surgical staples. After surgery, the animals were injected with 0.05 mL/kg of cefazolin (Covetrus Inc., Portland, ME, USA) and 0.15 mL/kg of sustained-release buprenorphine (Zoopharm, Wedgewood Pharmacy Inc., Laramie, WY, USA). A follow-up injection of buprenorphine with the same dosage was administered 72 h after surgery. Animals were allowed to recover for 7 days following surgery before the start of recordings.

### 2.4. Neural Recordings

Data were collected from animals weekly starting approximately one week following surgery and continued until 16 weeks post-implantation. Animals were anesthetized using 2.5% isoflurane supplemented with pure oxygen. After unconsciousness was confirmed using a toe pinch, the isoflurane level was reduced to 1.5–2.0% and data were collected using an Omniplex recording system (Plexon Inc., Dallas, TX, USA) at a sampling rate of 40 kHz. Wideband recordings of spontaneous activity in the motor cortex were collected for ten minutes from all 16 channels simultaneously.

### 2.5. Data Processing

The continuous wideband data were analyzed using the Plexon Offline Sorter software (x64 V4.7.1, Plexon Inc., Dallas, TX, USA). First, the wideband recording was processed using a 4-pole Butterworth bandpass filter from 300 to 3000 Hz. Next, channels were normalized to the median signal of all channels to reduce any effects from artifacts or noise [38]. Waveforms potentially associated with single units were identified as those deviating more than −4 standard deviations from baseline. Single units were automatically identified using a k-means scan to identify clusters of waveforms within the principal component space and verified manually. Figure 2 shows a representative filtered neural recording, raster plot, and the waveforms associated with two single units. The percentage of functional channels that recorded at least one unit in a session was calculated to obtain the active electrode yield (AEY). Signal-to-noise ratio was determined by dividing the peak-to-peak voltage (Vpp) of the single units by the noise level. The noise level was calculated by first excluding any sections of the filtered continuous signal linked to a single unit and then computing the root mean square (RMS) of the remaining signal.

To examine whether there were similar patterns of performance with respect to time across the arrays, hierarchical clustering analysis was performed to group the recording devices into two groups based on the changes in their AEY over the 16-week implantation period. Hierarchical clustering is a form of unsupervised learning that partitions unlabeled data based on their similarity. It considers each datum as a cluster, and then combines clusters based on similarity until a single cluster is reached. We determined similarity based on Euclidean distance between clusters [28]. This allows us to observe overall trends in the performance of the two types of devices, even when a minority of individual devices do not follow the overall trend. The weekly AEY measures for each array were processed using Origin Pro for hierarchical clustering.

In addition, we wanted to identify the ability of individual channels to record from the same single unit over time. We identified the subset of channels that showed exactly one single unit one-week post-implantation (silicon-based probes: *n* = 29; a-SiC devices: *n* = 44). Although these may not always be the same single unit, we believe it is a reasonable method of assessing the performance of devices in this area. We followed the single-unit channels each week for as long as a single unit was observed on the channel. We noted the voltage peak-to-peak on the first and last weeks and the length of time over which the unit could be detected. We calculated the percent decrease of peak-to-peak voltage and the average lifetime of a unit, calculated as the average number of weeks that each electrode site had units distinguishable from noise for each MEA type.

### 2.6. Immunohistochemistry

Immunostaining of rat brains implanted with MEAs was conducted as previously described [21]. In brief, rats were euthanized 16 weeks post-implantation via intraperitoneal injection of sodium pentobarbital (Virbac Corporation, Westlake, TX, USA) and transcardially perfused with 1× PBS followed by 4% paraformaldehyde (PFA) (Sigma-Aldrich, St. Louis, MO, USA). The skulls were then submerged in 4% PFA for 48 h prior to extraction of the MEAs. Brains were extracted and stored in PBS and sodium azide for 24 h before initiating the histological process. Brain tissue was embedded in agarose gel, sliced into 100 µm sections, and stored in PBS and sodium azide at 4 °C overnight. The immunohistochemistry steps were as follows:

Day 1: Slices were treated with sodium borohydride (Sigma-Aldrich, MO, USA) to quench autofluorescence; blocked and permeabilized in a solution containing 4% normal goat serum (Abcam Inc., Waltham, MA, USA), 0.3% Triton X-100 (Sigma-Aldrich, MO, USA), 0.1% sodium azide (Sigma-Aldrich, MO, USA) in PBS, and 2% bovine serum albumin (Sigma-Aldrich, MO, USA); then, they were treated with Image-iT (Thermo Fisher Scientific, Waltham, MA, USA) followed by primary antibody treatment and stored at 4 °C overnight.

Day 2: Slices were washed with PBS and Triton X-100 and incubated in the secondary antibody solution for 2 h (Abcam Inc., MA, USA). Tissue samples were then mounted on glass slides and covered with coverslips. The antibodies used are summarized in Table 2.

### 2.7. Image Analysis

Image acquisition and analysis was performed using previously described protocols [21,25]. In brief, image acquisition was performed using Nikon NIS Elements software (4.40.00 (Build 1084), Nikon Instruments, USA). For each slice, the maximum intensity projection image was captured over 35 µm thickness of the tissue using a 10× objective, over a 1300 × 1300 µm viewing window. All the capture parameters were conserved across all images. The GFAP intensity was calculated using a MATLAB code. Then, the signal intensity was captured in a 50 × 100 µm rectangular region starting from the edge of each implant hole, extending up to 500 µm away from the holes. The given value for each region was then normalized to the intensity value in the 450–500 µm region. Finally, the normalized values of all the slices were averaged across superficial (100–800 µm) and deep regions (800–1200 µm).

NeuN analysis was performed by manual counting of the neuronal cell bodies in the same 50 × 100 µm rectangular region up to 500 µm away from the identified implant holes. Cell counts in each region were normalized to the region 450–500 µm away from the identified implant holes and averaged over superficial and deep layers. In total, seven a-SiC implanted brains and three silicon-based implanted brains were used for histological analysis. The number of slices processed at each depth is summarized in Table 3. Data for a-SiC devices were previously reported in [21], while the present work introduces comparative immunohistology from implanted silicon-based probes for comparison.

Some slices were excluded from the analysis due to staining inconsistencies or tissue tears to prevent the introduction of artifacts that could compromise data integrity.

### 2.8. Statistical Analysis

The AEY one-week post-implantation was compared between the two groups using a test of proportions. Normality of the percent change in single unit Vpp and lifetime was assessed using Shapiro–Wilk’s test, with data being non-normal. The average AEYs of the groups of MEAs identified by clustering analysis were compared using a Mann–Whitney test. The profiles of Vpp, noise, SNR, and spike rate over time were assessed and compared between groups using simple linear regression.

Normality of the histological data was assessed using Shapiro–Wilk’s test, with all data confirming normal distribution. We then performed a two-way ANOVA to compare the effect of distance from the implantation holes on GFAP intensity and neuronal density for a-SiC and silicon-based probes. This was followed by a Tukey’s post hoc test to identify the distances at which there is no statistical difference between the a-SiC and silicon-based probe histology. Data are presented as the mean ± standard error of the mean. Statistical analysis was performed using GraphPad Prism (10.4.1 (627), Dotmatics, Boston, MA, USA).

## 3. Results

### 3.1. Neural Recording

As shown in Figure 3, a-SiC MEAs showed less decline in AEY compared to silicon-based MEAs in both acute and chronic neural recordings. One-week post-implantation, the AEY of the a-SiC MEAs (87.5%) was significantly greater than the AEY of silicon-based MEAs (71.3%) (*p* < 0.0001). The slope of the regression line of AEY versus week of implantation for the silicon-based MEAs (−3.8%/week) was significantly greater than the slope of the regression line of a-SiC MEAs (−2.1%/week) (*p* = 0.006). Six weeks post-implantation, the AEY of silicon-based MEAs permanently fell below 50%. a-SiC MEAs showed an AEY greater than 50% until fifteen weeks post-implantation.

We also examined the Vpp (Figure 3B), noise, SNR, and spike rate of single units recorded on both array types. We observed a significant difference in Vpp and noise over time. Overall, the Vpp and noise observed with a-SiC MEAs showed more stability throughout the study than silicon-based MEAs, although a-SiC MEAs did show a decline in Vpp (*p* < 0.0001, slope = −1.49 μV/week, 95% CI = (−2.05, −0.92)). The silicon-based MEAs showed a decline in Vpp (*p* < 0.0001, slope = −3.81, 95% CI = (−4.83, −2.80)) that was significantly greater than that seen in a-SiC MEAs (*p* < 0.0001). A similar trend was observed in the magnitude of noise. The a-SiC MEAs showed a small increase in noise over time (*p* = 0.0476, slope = 0.037 μV/week, 95% CI = (0.0004, 0.074)), while the silicon MEAs showed a significant decrease in noise (*p* < 0.0001, slope = −0.23 μV/week, 95% CI = (−0.28, −0.19)). The signal-to-noise ratio (SNR), which is calculated by dividing the Vpp by the noise level, showed some decrease in both a-SiC (*p* < 0.0001, slope = −0.175/week, CI = (−0.2327, −0.1172) and silicon MEAs (*p* = 0.011, slope = −0.1799/week, CI = (−0.3177, −0.04207)) but no significant difference in the rate of decline (0 = 0.9408). The spike rate of units recorded by a-SiC MEAs showed a small decline over time (*p* < 0.0001, slope = −0.40 Hz/week, 95% CI = (−0.51, −0.30)), whereas the spike rate of units recorded by silicon-based MEAs did not change significantly throughout the study (*p* = 0.9183, slope = 0. 0091 Hz/week, 95% CI = (−0.17, 0.18)).

Cluster analysis identified more stable recording clusters for a-SiC MEAs compared to the silicon-based arrays, reflecting the previously noted slower AEY decline in a-SiC MEAs (−2.1%/week) versus silicon-based MEAs (−3.8%/week). However, for both groups, there was a small number of counterexamples to these trends. Therefore, we compared the AEY profile over time for each MEA irrespective of type to identify any prominent modes or clusters. Figure 4A shows that hierarchical clustering applied to the AEY profiles yields two major groups that contain a mixture of both a-SiC and silicon-based array data. Two statistically distinguishable clusters could be identified based on AEY versus time for all MEAs (Figure 4B): (1) a rapid-decline group comprising six members with two a-SiC MEAs; (2) a slow-decline group comprising six members with five a-SiC MEAs. We then asked if the AEY between the two clusters was significant each week using a Mann–Whitney test. By 8 weeks post-implantation, there was a significant separation in the two groups that was observed for the remainder of the study (*p* < 0.01). These data suggest that both MEA types are capable of slow decline performance; however, the a-SiC devices are more likely to show this behavior and preserve function for extended periods of time.

We also examined the changes in individual units. As shown in Figure 5, a unit observed on an individual channel of an a-SiC MEA experienced an average 1% decline in peak-to-peak voltage. The silicon-based MEA, on the other hand, experienced a 30% decline, showing that silicon-based MEAs experience a significantly greater decline than a-SiC MEAs (*p* = 0.0006).

We also observed that an average lifetime of channel’s detection of a unit on an a-SiC MEA is 6.3 weeks, while the average lifetime of detection of a unit on a silicon-based MEA is 2.4 weeks, indicating that an individual channel of an a-SiC MEA can record a single unit for significantly longer than a channel of a silicon-based MEA (*p* < 0.0001).

### 3.2. Immunohistochemistry

Figure 6 presents representative immunohistochemistry (IHC) images and corresponding quantitative analyses comparing the tissue response to a-SiC and silicon-based probes. Cortical slices from both superficial (100–800 µm) and deep (800–1200 µm) layers beneath the pial surface were stained for glial fibrillary acidic protein (GFAP) to identify activated astrocytes and for NeuN to mark neuronal nuclei. These two depth ranges were selected to explore the impact of implantation depth on tissue response: the superficial layers where no electrode sites reside, and the deep layers where the electrode sites are located. Figure 6a,b show GFAP staining in brain slices from the superficial and deep cortical layers, respectively, with the a-SiC implanted slice on the left and the silicon implanted slice on the right. Quantitative analysis of GFAP intensity (Figure 6c,d) indicated a marked difference in the astrocytic response between the two probe types. In the superficial layers, the a-SiC implants showed a lower astrocytic response compared to silicon-based probes, with the mean normalized GFAP intensity of 0–50 µm from the implant being 1.99 ± 0.30 (mean ± SEM, n = 17) for a-SiC compared to 5.56 ± 1.90 (n = 12) for silicon probes. The intensity dropped to 1.04 ± 0.01 (n = 17) at 450–500 µm for a-SiC and 1.25 ± 0.20 (n = 12) for silicon-based probes. A two-way ANOVA with Tukey’s post hoc test revealed a statistically significant difference in GFAP expression between the probe types up to 150 µm from the implant site (*p* ≤ 0.001). At distances beyond 150 µm, no significant differences were observed (*p* = 0.10).

In the deep cortical layers, the astrocytic response followed a similar trend, with a-SiC implants inducing less GFAP expression compared to the silicon-based probes. The mean normalized GFAP intensity at 0–50 µm from the implant site was 1.48 ± 0.16 (n = 18) for a-SiC and 6.10 ± 2.0 (n = 9) for silicon-based probes. Statistically significant differences in GFAP intensity between the two probes were observed up to 300 µm away from the implant (*p* < 0.001), but no significant differences were found beyond that distance (*p* = 0.10).

Figure 6e–j display NeuN-stained slices from superficial and deep cortical layers, along with the corresponding quantification of neuronal density (Figure 6g,h). In the superficial layers, a significant reduction in neuronal density near the implantation site (0–50 µm) was observed for brains implanted with silicon-based probes compared to a-SiC probes (*p* = 0.04). The mean normalized neuronal cell count at 0–50 µm was 0.92 ± 0.05 (n = 17) for a-SiC and 0.78 ± 0.004 (n = 11) for silicon-based probes. At distances further from the implant (450–500 µm), the neuronal density was comparable between the two probe types.

In the deep cortical layers, the neuronal cell count did not show significant differences between the a-SiC and silicon-based probes, with the mean normalized cell count at 0–50 µm being 0.95 ± 0.05 (n = 18) for the a-SiC and 0.85 ± 0.07 (n = 9) for the silicon-based probes. Both probe types showed similar values at the 450–500 µm distance.

The results suggest that a-SiC probes, with their smaller cross-sectional area, cause significantly less astrocytic activation and better preserve neuronal density near the implant site compared to silicon-based probes, which suggests that a-SiC probes may minimize tissue damage and help maintain healthier surrounding tissue over time. The reduced GFAP expression in proximity to a-SiC implants likely contributes to their ability to support improved long-term recording stability. Conversely, the higher GFAP expression observed with the silicon-based probes, particularly at electrode recording depths, could contribute to a reduction in chronic recording performance, as astrocytic scarring and inflammation are believed to impair electrode functionality over time [39]. Overall, the findings highlight the potential of a-SiC probes to mitigate immune response and preserve neuronal architecture, offering a promising avenue for improving the longevity of neural implants.

## 4. Discussion

Our previous study [21] detailed the neural recording capability and tissue response to a-SiC implants. In this study, we have provided a comprehensive comparison to implanted silicon-based probes, showing that a-SiC MEAs with a reduced cross-sectional area record neural activity with greater stability than silicon-based MEAs of similar architecture but with larger shanks. At 16 weeks post-implantation, a-SiC MEAs show single unit activity on 44.1% of channels, while silicon-based MEAs show single unit activity on just 15.0% of channels, consistent with previously reported results from similar multi-shank silicon-based MEAs [40,41]. Interestingly, work in our laboratory comparing the AEY from single-shank silicon-based MEAs shows more apparent stability compared to multi-shank silicon-based MEAs [41].

In addition, we observed significantly more stability in the Vpp of single units recorded on a-SiC MEAs compared to silicon-based MEAs. In conventional MEAs, we also observed a significant decline in noise over time in recordings. Between the two probe types, we did not observe a significant difference in the decline of SNR over time. The decline in noise observed in silicon-based MEAs may be related to the greater tissue response we observed. Increased scarring around the recording site creates distance from biological sources of noise, resulting in lower recorded amplitude. It may also be related to the methods used to analyze the neural recordings. Single units were identified based on their separation from the noise. Because of this, as the Vpp of single units declines, they will only be detectable if the noise is also low, resulting in the analysis drawing primarily from channels where less noise was observed.

We have also compared the performance of individual MEAs using hierarchical clustering to separate their performance as either rapid-declining or slow-declining. The majority of a-SiC MEAs were classified as slow-declining, while the majority of silicon-based MEAs were classified as rapid-declining. This suggests that a-SiC MEAs, due to their smaller cross-sectional area, are more likely to show slow decline in performance and preserve function over time. The deviation of some devices from the overall trend may be partially attributed to the fabrication process of the a-SiC devices, which were produced in-house. Unlike commercially available MEAs, these devices are more susceptible to structural damage. For example, repeated connections and disconnections of the device and recording system during recording sessions could damage channel traces. We also observed that units observed on individual channels of silicon-based MEAs on average showed significant loss in amplitude over the course of the study and were detectable, on average, for less than 3 weeks. Units observed on individual channels of a-SiC MEAs, on average, showed little or no change in amplitude and were detectable for over 6 weeks, on average. This indicates that the a-SiC MEAs show single unit activity significantly longer and at a higher amplitude than the silicon-based MEAs.

Histological analysis confirmed that a-SiC-based probes induce a significantly lower tissue response compared to the silicon-based probes, with differences observed at varying depths. Although the intrinsic layer-specific properties of astrocytes may play a role [42,43], the increased flexibility of a-SiC probes at deeper cortical regions likely contributed to these depth-dependent differences, allowing for better tissue integration, aligning with previous studies [21,44,45,46]. There are a variety of possible causes of immune response to microelectrode arrays, including blood–brain barrier disruption, inflammation, and reactive oxidative species. Studying the impact of specific mechanisms of immune response, and its relationship to different devices, could be the subject of a larger and more comprehensive study [47].

The material properties of the probes further support these observations. Using a Young’s modulus of 75 GPa, as employed previously [32], the corresponding stiffness of the a-SiC probes is 24 mN/m. In comparison, silicon-based probes have a Young’s modulus of approximately 190 GPa [33,48], yielding a stiffness of 900 mN/m. A meta-analysis by Stiller et al. [33] highlighted a strong correlation between device stiffness and the severity of the immune response. These findings suggest that using more flexible probes should reduce tissue responses, likely by reducing stiffness mismatches with the brain and alleviating mechanical strain on surrounding tissue.

To further improve tissue integration, ultra-flexible neural devices such as nanoelectronic threads (NETs) have been developed, with cross-sectional areas as small as 10 × 1.5 µm^2^. These dimensions drastically lower stiffness, enhancing biocompatibility and neural integration. However, these flexible devices require rigid insertion shuttles for implantation [26], which can increase acute tissue damage [48]. While ultra-flexible devices offer superior biocompatibility and long-term recording stability compared to rigid devices, their extreme flexibility presents a more challenging surgical implantation [26].

Furthermore, a-SiC probes maintained neuronal density across all cortical layers near the implant site, with significantly higher neuronal cell body counts observed within 50 µm of the probe in superficial layers compared to silicon-based MEAs. This finding aligns with prior studies reporting a loss of neurons in close proximity to silicon implants 16 weeks post-implantation [16]. The results highlight a clear association between higher levels of activated microglia and reactive astrocytes and increased neuronal loss, emphasizing the importance of reducing chronic inflammation to preserve neuronal integrity around implanted devices. While the tissue implanted with a-SiC devices showed no significant reduction in neuron count near the device at greater depths, the neural recordings did show a decrease in activity. This may be due to a loss of functional connectivity between neurons in different cortical layers [49], or significant structure and functionality impairments affecting the activity of individual neurons [50], and not directly the result of neuron loss.

It remains to be demonstrated that a-SiC MEAs can be consistently implanted into large animal models, particularly non-human primates. There are currently unpublished data, including single units, recorded acutely from a small number of pigs using a-SiC MEAs. However, it remains to be studied how the devices will perform chronically and whether they can be implanted consistently in a larger and more comprehensive study.

Our findings are largely consistent with prior work demonstrating that devices of reduced-cross-sectional area show more stable chronic recording performance and reduced tissue response [26]. Overall, these findings underscore the advantages of a-SiC MEAs in both minimizing tissue damage and improving long-term neural recording stability, and highlight the potential of a-SiC MEAs for future neural interface applications.

## 5. Conclusions

In this work, we demonstrated that a-SiC MEAs can record single units with greater stability and longevity than a silicon-based MEA over a 16-week chronic study. The recordings maintain higher AEY while individual channels also show more stability in peak-to-peak voltage and the presence of detectable single units. The improved recording outcomes are supported by histological analysis and exhibited a lower tissue response to the implanted a-SiC probes in the cortex compared to the silicon-based probes.

The results suggest that a-SiC probes, with their smaller cross-sectional area, induce less astrocytic activation and better preserve neuronal density near the implant site. This reduced tissue response likely contributes to the improved long-term recording stability observed with a-SiC MEAs. In contrast, silicon-based probes, with their larger cross-sectional shanks, showed higher GFAP expression, particularly at electrode recording depths, which may impair electrode functionality over time due to astrocytic scarring and inflammation.

Overall, a-SiC MEAs present a promising platform for chronic neural recording, offering enhanced single-unit recording performance, reduced immune response, and better preservation of the surrounding neuronal architecture. These findings underscore the potential of a-SiC probes to improve the longevity and functionality of neural implants.

## Figures and Tables

**Figure 1 micromachines-16-00113-f001:**
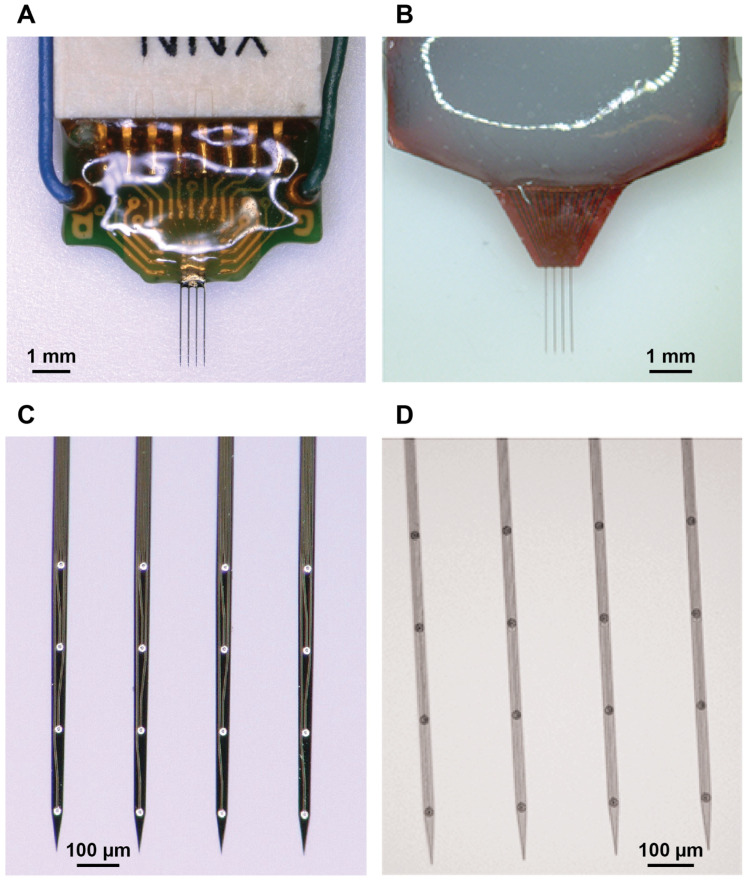
Representative images of implanted MEAs. (**A**) Image of silicon-based MEA. (**B**) Image of an assembled a-SiC MEA. (**C**) Zoomed image of the shanks and electrode sites of a silicon-based MEA. (**D**) Zoomed-in image of the shanks and electrode sites of an a-SiC MEA.

**Figure 2 micromachines-16-00113-f002:**
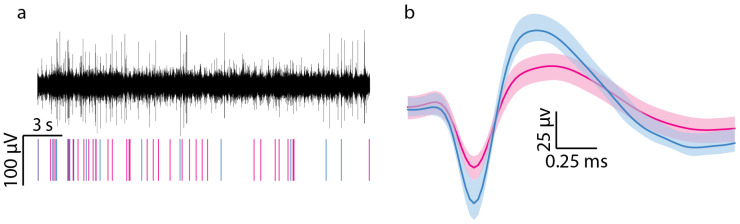
Process of identifying a representative single unit. (**a**) Filtered neural recording and raster plot showing spiking activity associated with two single units, represented in blue and pink. (**b**) The two single units identified from the neural recording shown in (**a**), with the average waveforms of each unit shown in bold.

**Figure 3 micromachines-16-00113-f003:**
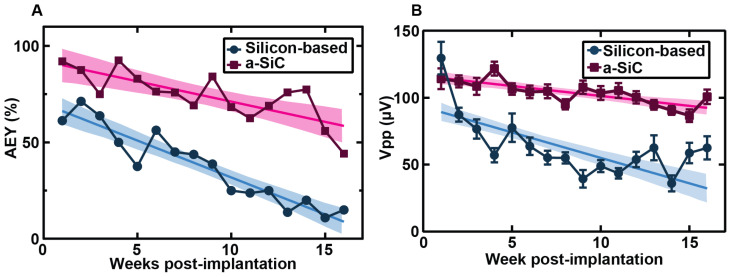
Comparison of silicon-based and a-SiC MEA performance. Data from a-SiC arrays were collected from our prior work [21]. (**A**) Comparison between the AEY of silicon-based (blue circles) and a-SiC (pink squares) MEAs weekly over the 16-week chronic study with 95% confidence intervals and regression line. At week one, the AEY are significantly different (*p* < 0.0001). (**B**) Comparison between silicon-based and a-SiC MEAs’ peak to peak voltages (Vpp) with SEM, 95% confidence intervals, and regression line.

**Figure 4 micromachines-16-00113-f004:**
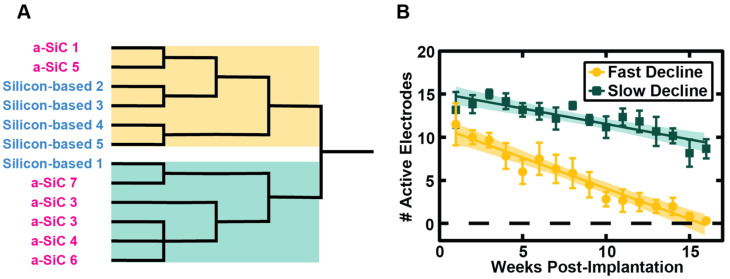
Cluster comparison between the AEY of slow-declining and rapid-declining MEAs. (**A**) An unsupervised dendrogram of the clustering process, with the rapid-declining group on top in yellow and the slow-declining group on the bottom in green. (**B**) The average number (#) of electrodes recording single units per animal in each mixed cluster over the course of the study, with error bars showing the standard error of the mean (SEM).

**Figure 5 micromachines-16-00113-f005:**
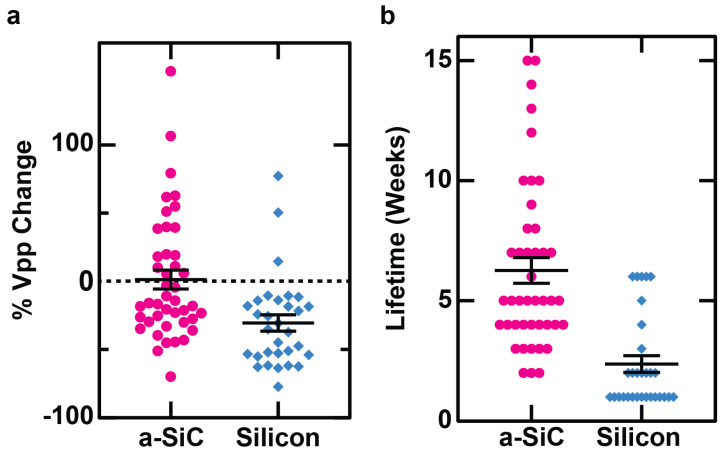
Individual channel activity over detection period of single units. (**a**) The percent change in peak-to-peak voltage between the first and last week on which each unit could be identified. (**b**) The average duration for which a unit could be identified on a given electrode channel.

**Figure 6 micromachines-16-00113-f006:**
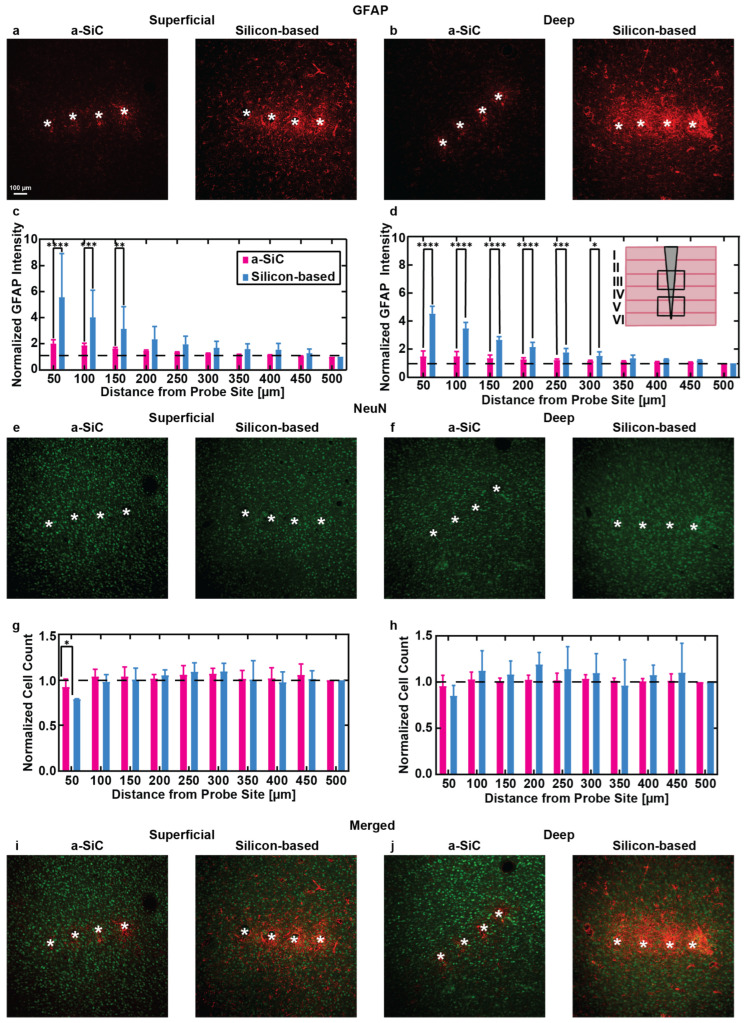
(**a**,**b**) Representative images of brain sections stained for glial fibrillary acidic protein (GFAP), showing immune responses at superficial (100–800 µm) and deep (800–1200 µm) depths below the pial surface for a-SiC and silicon-based probes. (**c**,**d**) Normalized GFAP intensity (mean ± SEM) as a function of distance from the probe site, comparing the immune response to a-SiC and silicon-based probes (mean ± SEM). The inset diagram in (**d**) demonstrates the implant location relative to the cortical layer. (**e**,**f**) Brain sections stained for neuronal nuclei (NeuN) in superficial and deep layers near both probe types. (**g**,**h**) Normalized NeuN cell count at increasing distances from the probe sites, comparing neuron density between a_SiC probes (pink) and silicon-based probes (Blue). (**i**,**j**) Merged images of GFAP and NeuN staining in superficial and deep brain layers for both a-SiC and silicon-based probes. White asterisks represent the identified implant holes. The statistical significance is represented by black asterisks (* *p* < 0.05, ** *p* < 0.01, *** *p* < 0.001, **** *p* < 0.0001).

**Table 1 micromachines-16-00113-t001:** Parameters used for flexural rigidity and stiffness calculations.

Probe Type	Young’s Modulus (E)	Width (b)	Height (h)	Shank Length (L)
a-SiC	75 [32]	20 µm	8 µm	2 mm
Silicon-based	190 [33]	45 µm	15 µm	2 mm

**Table 2 micromachines-16-00113-t002:** Antibodies used for immunohistochemistry.

Antibody	Target	Concentration	Supplier	Catalog Number
NeuN	Neuronal Nuclei	1:500	Abcam	ab4674
GFAP	Astrocytes	1:500	Abcam	ab104224
Alexa Fluor 555	NeuN	1:4000	Abcam	ab150118
Alexa Fluor 647	GFAP	1:4000	Abcam	ab150171

**Table 3 micromachines-16-00113-t003:** Sample size for MEA implanted brains used for immunohistochemistry.

	a-SiC (n = 7)	Silicon-Based (n = 3)
Superficial (100–800 µm)	17 slices stained with GFAP and NeuN	GFAP: 12 slices NeuN: 11 Slices
Deep (800–1200 µm)	18 slices stained with GFAP and NeuN	9 slices stained with GFAP and NeuN
	a-SiC (n = 7)	Silicon-based (n = 3)

## Data Availability

Data available upon reasonable request.

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
