# Peer review of "Enhanced Performance of Novel Amorphous Silicon Carbide Microelectrode Arrays in Rat Motor Cortex"

_micromachines, 2025, doi:10.3390/mi16020113_

Round 1

Reviewer 1 Report

Comments and Suggestions for Authors

The manuscript describes the performance comparison of silicon-based devices and low flexural 18 rigidity amorphous silicon carbide (a-SiC) MEAs for long-term recording from motor cortex of female Sprague-Dawley rats. The topic is interesting. The results also support the conclusions. However, some points still need to be addressed before it is suitable for publication.

1. Why the motor cortex was selected for recording? Why female SD rats were used?

2. What is the advantages of a-SiC MEAs compared to the commercial available MEAs?

3. It is not clear the underlying mechanism of reduced immune responses of a-SiC MEAs? More characterization and analysis on the interface of microelectrode and brain tissues should be provided.

Author Response

Comment 1: Why the motor cortex was selected for recording? Why female SD rats were used?

Response 1: The motor cortex was selected for recording because it is a critical region involved in motor control, making it highly relevant for studying the performance of MEAs in applications such as motor prostheses. It also exhibits neural activity under anesthesia.

We have included this point in page 4, lines 142 - 144

Sprague-Dawley (SD) rats were chosen because they are a well-established model widely used in research, including motor cortex implants, as evidenced by existing studies. Their relatively large brain size facilitates precise implantation and reliable neural recordings from specific cortical areas. Additionally, female SD rats were used to minimize variability in experimental results, as male rats may exhibit greater aggression and hormonal fluctuations that could affect outcomes.

These points are added to page 4, lines 128-132.

Comment 2: What is the advantages of a-SiC MEAs compared to the commercial available MEAs?

Response 2: As demonstrated in our results, a-SiC MEAs exhibit enhanced recording stability and longevity compared to commercially available silicon-based devices. This finding is supported by histological analysis, which shows a reduced foreign body response (FBR) in brain tissue implanted with a-SiC MEAs. These results suggest that the reduced cross-sectional area of the a-SiC MEAs minimizes tissue response, thereby enhancing the recording performance of a-SiC probes relative to silicon-based probes.

These points have been summarized in the conclusions section, page 13, lines 482-487.

Comment 3: It is not clear the underlying mechanism of reduced immune responses of a-SiC MEAs? More characterization and analysis on the interface of microelectrode and brain tissues should be provided.

Response 3

There are a variety of possible causes of immune response in microelectrode arrays, including blood-brain barrier disruption, inflammation, and reactive oxidative species. Studying the impact of specific mechanisms of immune response, and its relationship to different devices, could be the subject of a larger and more comprehensive study. We added this text to page 12, lines 438 - 442.

Reviewer 2 Report

Comments and Suggestions for Authors

    While the manuscript is detailed and thorough, there is some redundancy in the text (e.g., repeated emphasis on FBR reduction). Refining such points can enhance readability without losing any impact. Also, it is suggested to include a brief discussion of the existing challenges in adopting advanced neural implants.

·     The technical details are well-covered; however, it is suggested to expand the practical implications for broader neural engineering applications (e.g., brain-machine interfaces or prosthetic control).

·   There are some errors in the manuscript that needs to be rectified. For instance,  reference no. 26 is incomplete and lacks a valid URL. It is suggested to provide a full citation or remove if unavailable.

·     In Figure 6 (panels c and d), the x-axis indicates "Distance from Probe Site (µm)" but lacks a note about whether these distances are averages or specific slices. It is better to clarify in the legend whether the distances represent averages across slices or specific points from the implant site. The GFAP and NeuN quantifications are thorough but is it possible to add a small inset diagram showing the implant location relative to the cortical layers for better anatomical context?

·   The variability in AEY decline among the clusters is well discussed. Is there any explanation for the observed clustering patterns, such as manufacturing differences or animal-specific factors?

Author Response

Comment 1: While the manuscript is detailed and thorough, there is some redundancy in the text (e.g., repeated emphasis on FBR reduction). Refining such points can enhance readability without losing any impact. Also, it is suggested to include a brief discussion of the existing challenges in adopting advanced neural implants.

Response 1: Thank you for your comment. We have edited the manuscript to avoid redundancy.

It remains to be demonstrated that a-SiC MEAs can be consistently implanted into large animal models, particularly non-human primates. There is currently unpublished data, including single units, recorded acutely from pigs using a-SiC MEAs. However, it remains to be studied how the devices will perform chronically, and whether they can be implanted consistently in a larger and more comprehensive study.

These points are added to page 13, lines 471 - 475.

Comment 2: The technical details are well-covered; however, it is suggested to expand the practical implications for broader neural engineering applications (e.g., brain-machine interfaces or prosthetic control).

Response 2: Thank you for your suggestion. We added: “Among other applications, brain-machine interfaces can be used to restore limb function after paralysis caused by spinal injury or allow the control of prosthetic limbs by amputees.”

The text is added to page 1, lines 39 - 41.

Comment 3: There are some errors in the manuscript that need to be rectified. For instance, 

reference no. 26 is incomplete and lacks a valid URL. It is suggested to provide a full citation or remove if unavailable.

Response 3: Thank you for bringing this to our attention. The error has been fixed.

Comment 4:   In Figure 6 (panels c and d), the x-axis indicates "Distance from Probe Site (µm)" but lacks a note about whether these distances are averages or specific slices. It is better to clarify in the legend whether the distances represent averages across slices or specific points from the implant site. The GFAP and NeuN quantifications are thorough but is it possible to add a small inset diagram showing the implant location relative to the cortical layers for better anatomical context?

Response 4: Thank you for your comment. We revised the caption to “Normalized GFAP intensity (mean ± SEM) as a function of measured distance from the probe site in each slice, with intensity values averaged across slices at each distance, comparing the immune response to a-SiC and silicon-based probes”.

Additionally, a cartoon of the general region of implant is added to figure 6d.

Comment 5: The variability in AEY decline among the clusters is well discussed. Is there any explanation for the observed clustering patterns, such as manufacturing differences or animal-specific factors?

Response 5: The clustering patterns in AEY decline may be partially attributed to the fabrication process of the a-SiC devices, which were produced in-house at the university. Unlike commercially available MEAs, these devices are more structurally susceptible to damage. For instance, repeated connections and disconnections of the headstage to the bond pad during recording sessions could potentially damage channel traces, leading to variability in device performance and clustering patterns.

These points are added to page 12, lines 422 - 426.